# Implementation of the Lean Healthcare System in the Emergency Room of the Clinical Hospital of the Federal University of Uberlândia: A Case Study

**DOI:** 10.3390/ijerph20247184

**Published:** 2023-12-15

**Authors:** Paulo Sergio de Freitas, Guilherme Silva de Mendonça, Elmiro Santos Resende

**Affiliations:** Faculty of Medicine, Federal University of Uberlândia, Patos de Minas 38700-002, Brazil; guilhermesilvamendonca@gmail.com (G.S.d.M.); esr_udi@hotmail.com (E.S.R.)

**Keywords:** lean in emergencies, emergency room overcrowding, lean health, management of health services, hospital management

## Abstract

The objective of this study was to analyze the effectiveness of the implementation of the lean healthcare system at the emergency room of the Clinical Hospital of the Federal University of Uberlândia, based on a comparison of hospital indicators obtained over the three phases corresponding to the period of one year before the implementation (T1), the year during the implementation (T2) and one year after implementation (T3). The methodology applied through this study can be classified as a case study that is exploratory and descriptive and developed in stages. Based herein on the search for hospital indicators, as occurred in the implementation of a lean process at the Clinical Hospital Emergency Department, along with a description of the implemented lean system. During the collection period of data relevant to the National Emergency Department Overcrowding Score and Length of Stay Indicator, the motivation of the teams grew, but with a notable tension between municipal management and hospital management. It was found that, despite the fluctuations, the patient length of stay in the Emergency Room remained high. With the exception of the variable of female deaths before 24 h of hospitalization, all other variables showed percentage increases before and after the intervention. This study reported the difficulties encountered by HC-UFU in implementing the lean project in an emergency room, thus ensuring that other institutions that intend to implement this project do not make the same types of mistakes.

## 1. Introduction

The Brazilian public health system suffers from a lack of allocation of resources, and this is considered one of the major concerns in the lives of Brazilians. However, specialists affirm that this problem is not based only on finances, but also on the lack of qualified management in the health system [1]. According to data published by the Federal Court of Accounts (TCU), 64% of hospitals are always overcrowded, while only 6% are not. In the analysis carried out by the TCU, the allocation of patients to corridors of emergency hospitals represents a problem at 47% of the hospitals evaluated, whereas in 33%, this fact is always present, and is frequently reported in 14% of hospitals [1]. The TCU also pointed out other situations encountered in overcrowded Brazilian emergency rooms, those being patients lying on benches, insufficient space between beds, patients on stretchers in the reception area, patient identification fixed on walls or stretchers, beds numbered in corridors, bodies curled up on the floor of the emergency room, patients with long waiting times for a place in the ICU, emergency room with low levels of bed rotation and problems in referring patients to other units [1].

In light of this situation, it becomes necessary to understand that the standing of the modern hospital is that of a complex organization, which should be capable of offering quality services and safety to an ever more informed and demanding public. In addition, it should take care of its very own sustainability, as it is faced with a competitive market that is on many occasions held down by laws and demands from professional associations, in addition to having to constantly be at the forefront of technological advances. In order for this to occur in an adequate manner, it is necessary to set a delicate balance conducive to the development of hospital actions, which itself is a fact that depends on the management model being directed toward new tendencies. However, consideration should always be given to the proper relationship between cost and efficiency, in addition to always focusing on the quality of care and patient safety. To reach the goal of transforming the hospital into a competitive company, the hospital administration needs to focus on maintaining its professionalism, along with the adoption of an adequate production model [2].

Due to the need for a more efficient management system with ever-limited resources, some tools, commonly adopted in manufacturing, have been adapted to the health system, among these, one highlights that known as “lean production (LP)”, which is recognized under the term of lean health [3]. This tool is a management method that allows, using the simple steps that should be constantly observed and implemented, for the identification of what generates value for the patient, as well as the points where waste occurs along a production chain. These waste points can, as such, be identified, minimized, or even eliminated. Those resources that were wasted in the past are thus redirected to procedural steps that aggregate the value of the service being offered [4]. The first publications on the lean healthcare method occurred in 2002 [5] and, since then, it has been implemented in a number of clinics and hospitals across various countries. Successful cases are described in the literature with significant gains across numerous management parameters, as in the reduction of serious hospital complications, elimination of waste in various comportments, reduction in the waiting for care times, increases in quality of the service provided and standardization of documentation and medical records, which can make hospital activities safer and more efficient [4].

From the point of view of Mintzberg [6], the application of the lean healthcare philosophy has spread through various countries, since the beginning of the 2000s, using research based on the compilations associated with different cases, themes, and approaches in health establishments. However, in Brazil, the use of such systems is still a relatively recent phenomenon [7]. In this context, lean philosophy attracts many researchers who base its use and exploration on its original concepts in lean manufacturing, where these have been transferred directly to the field of health [8]. These studies, however, show that lean tools are generally applied to specific areas or processes, and not directly to the culture of the organization [9,10].

The problem of access and quality in the emergency care provided reflects the state of an expanding health system, which still has not found the balance between population needs and the quality of the existing offer. As such, one notes substantial injustices in the access to opportunities, quality services and assistance results for patients and family members, as well as in the working conditions of the health teams [11].

The Emergency Medical Service (EMS) represents an essential component in the health system, as it provides immediate access to evaluative care, stabilization, and treatment of patients with acute conditions that involve traumatic and non-traumatic pathologies. The EMS has become an entrance point used in an inadequate fashion by part of the population, creating an entrance point where a majority of patients are introduced to the health system. At the same time, it is also an opportune place to advance the reforms needed to ensure universal access to essential healthcare services [12].

The Clinical Hospital of the Federal University of Uberlândia (HC-UFU) is not different from the system described above. As this hospital has a history of occupation above 100%, with subsequent low bed rotation, dissatisfaction of the participants involved concerning actions from the Public Ministry, resulting in negative coverage in the media that demonstrates the dissatisfaction of the population, thus leading to repercussions in terms of the internal difficulties in creating shifts, as well as the dissatisfaction of health professionals.

The Clinical Hospital of the Federal University of Uberlândia (HC-UFU) is a Public University Hospital that provides care exclusively through the Unified Health System (SUS). The HC-UFU accounts for the largest share of the public health network in Minas Gerais, occupying third place in the ranking of the largest university hospitals in the teaching network of the Ministry of Education (MEC). Sustaining an active physical area of more than 51.6 mil m^2^, HC-UFU is a reference in medium and high complexity for 86 municipalities from both macro and micro-regions of the Triângulo Norte, providing care to a population of approximately 2,906,791 inhabitants (source: SES/MG—Department of Integrated Conventional Programming). Originally constructed as a teaching unit of the professional assembly of the Medicine course at the former School of Medicine and Surgery of Uberlândia, HC-UFU was inaugurated on the 26 August 1970 and went on to initiate its activities in October of the same year, with only 27 beds [13].

Through the enactment of the 1988 Constitution, the HC-UFU became an important link in the SUS chain, mainly when it came to providing urgency and emergency along with high complexity care, where it is currently the only regional public hospital maintaining open access 24 h a day for all levels of health care [13].

The constant search for quality and reference care has led to the development of actions in the sense of offering a comprehensive and humanized care program, which adapts its provided care to meet the demands made by SUS. Therefore, a constant search exists for resources that guarantee efficiency and efficacy in service, while providing improvements in teaching, research, and care, thus guaranteeing quality in the services provided to the population that integrate actions in a participatory manner.

Through its importance and representativeness across the region, in its particularities of being a university care-providing hospital, combined with its problematic history of chronic overcrowding, lack of on-duty attendants, stretchers in the corridors, along with interventions by the Public Prosecutor’s Office and dissatisfaction of both health professionals and the population, led to the selection of HC-UFU for the implementation of lean in emergencies.

The HC-UFU Emergency Department currently has 92 beds, with 18 registered for Internal Medicine, 20 for General Surgery, 13 for Orthopedics, 12 for Gynecology and Obstetrics and 11 for Pediatrics. In addition, it has an eight-bed trauma room that serves as access for more severe patients and an emergency department with 10 beds. This infrastructure provides a support role for patients in need of intensive care [13].

The construction of the HCU-UFU emergency department occurred in 1980, and at that time the population of Uberlândia was 240,967 inhabitants. However, since then only refurbishments have been carried out, without the expansion of its physical space. Contrasted with this situation is the fact that today the city has a population that exceeds 700,000 inhabitants, according to an estimate by the IBGE for 2022. Even so, the HCU-UFU emergency room is still responsible for tertiary urgent and emergency care for the entire macro-region of Triângulo Mineiro and Alto Paranaíba; that is, it is a reference in hospital and outpatient care in medium and high complexity for the 86 municipalities in the state of Minas Gerais [13].

On 30 April 2018, the HCU-UFU launched an operational excellence program based on the lean methodology. The initiative from the Health Ministry, using Proadi-SUS, created the Strategic Action Support Project. Through this project, the Sírio-Libanês Hospital (Syrian Lebanese Hospital) in São Paulo, developed this program that had already been implemented in emergency wards of another six teaching hospitals in the SUS network [13]. 

Noteworthy here is that due to the turnover of directors and technical teams that occurred over the lean implementation period at the HC-UFU, many of the original records of the technical team that implemented the system were lost; hence, this is still reflected in the current collection of records. The research was performed based on general information obtained from the Statistics Sector of HC-UFU and, in part, on the recovery of records performed by the author of this study, who effectively participated in the implementation process of lean in emergencies at the referred hospital.

In light of the aforementioned, this present study had as its objective to analyze the effectiveness of the implementation of the lean healthcare system at the emergency department of the Clinical Hospital of the Federal University of Uberlândia (HC-UFU), through the use of a comparison of hospital indicators obtained from the three phases that correspond to the periods of one year before implementation (T1), the year during the implementation (T2), and one year after the implementation (T3).

## 2. Material and Methods

Applied research is characterized by the practical interest that its results be applied or used in an immediate fashion in the search for solutions to problems that occur in real life [14].

Therefore, this research is considered as applied, as it has the characteristic of contributing, through its results, to the very management of the hospital organization itself. This resonates with the object of this study, and to others of the same model, since the description of the implementation of the lean healthcare methodology in a large public hospital could serve as a model for other public health organizations to adopt the same strategy. In the same way, the description of the possible difficulties conceptualized during the process can be used as a learning regarding contingencies common to public organizations.

The research herein is classified as a case study and is aimed at deepening the lean theme and its application at HC-UFU, seeking to describe, in detail, the processes used in its implementation. A case study is characterized as a detailed study of one or a limited number of objects that seeks to obtain the maximum amount of information on the focal study theme [15].

The study herein is also classified as exploratory and descriptive. Exploratory, as it aims for greater familiarization with the data, through the use of a process that captures bibliographic portfolios. This familiarization should lead to greater clarity, raising of hypothesis and descriptiveness, as it aims at explaining the characteristics of a phenomenon or population, in relation to the research variables [15].

This study was developed using the following steps: the search for hospital indicators; following the implementation steps of lean in emergencies as at HC-UFU; and providing a description of the implemented lean system, as described below.

### 2.1. The Search for Hospital Indicators

The present research was performed from the collection of local data that refers to the experience of implementing lean healthcare, through comparisons of hospital records before (T1), during (T2) and after (T3) the implementation of the project.

The search for this information was performed in the Statistics Sector of HC-UFU, using the experimental method with exploratory analysis of the data referring to the emergency room of the HC-UFU, where a time series study was performed through the use of secondary data of the attendances that took place at the HC-UFU emergency room.

For the comparison between the periods before (T1), present (T2) and after (T3) the implementation of lean, hospital data were used, and the indicators that corresponded to the 10 variables were analyzed: Male and Female Admittance/Hospitalization; Male Discharge and Female Discharge; Male Transfer and Female Transfer, Male Death < 24 h and Female Death < 24 h; Male Death > 24 h and Female Death > 24 h, and general occupancy rate. All the raw data, referring to 36 months of evaluation, were supplied by the Statistics Sector of HC-UFU on spreadsheets without patient identification.

A time series study was carried out with secondary data on the attendances that occurred in the emergency room of the University Hospital of the Federal University of Uberlândia, Minas Gerais. For data processing, concerning the profile of attendances at the emergency room of the University Hospital, between April 2017 and March 2020, the decision was reached to divide this period into three intervals: Pre-intervention Stage (T1)—2017 (April 2017 to March 2018); Intervention Stage and implementation of the lean healthcare system (T2)—2018 (April 2018 to March 2019); and Post-intervention Stage (T3)—2019 (April 2019 to March 2020).

The data were analyzed using software R, version 4.2.1, a tool commonly used for analysis and manipulation of data, together with Excel. With the aim of describing the population in focus, a descriptive analysis was performed across all the variables related to care. Therefore, the cases were expressed in absolute and relative frequencies, and then the descriptive statistic was used with measures of central tendency and dispersion of each variable in each analyzed period. Following this, the Analysis of Variance (ANOVA) was performed to compare the parametric variables found along the three stages of the research, using the Tukey test as the post hoc measurement. 

For the trend analysis, standardized ratios by gender were calculated using population data as the standard population from the municipality of Uberlândia was found in the 2010 census and available on the DATASUS website. Next, trend analyses were performed for all variables separated by gender.

Trend analyses for time series were performed by linear regression using the Prais–Winsten method with robust variance. With the calculation of the β coefficient and the Standard Error (SE) obtained through the regression analysis, the Annual Percentage Change (APC) and respective 95% Confidence Interval (95%CI) were calculated. With the aforementioned at hand, one notes that the trends were stationary (*p* > 0.05), decreasing (*p* < 0.05 and negative regression coefficient) or ascending (*p* < 0.05 and positive regression coefficient), in each region of Minas Gerais, and stratified by gender and age group. 

For these analyses, the natural logarithmic transformation of the ratios was performed, which is capable of reducing the heterogeneity of the variance of the regression analysis residues. All data were considered significant with *p*-value < 0.05. 

Specific Lean in emergency scores (NEDOCS and LOS) were only recorded during the period that the implementation team was able to record them and were only used to demonstrate trends over time. As the evaluation sheets corresponding to pre- and post lean were not encountered, substitute indicators that were closest to these were adopted for comparison between the studied periods.

The indicators were compared one to the other for the three analyzed periods, with the study directed as follows: Previous Phase—Pre-intervention (T1), considered between the months of April 2017 to March 2018, completing 12 months before the implementation of lean; Intra-intervention Phase—contemporary (T2), considered between the months of April 2018 to March 2019, over a total of 12 months, which was the implementation period; and Post-intervention Phase (T3), successive, considered between the months of April 2019 to March 2020, completing 12 months after the Lean implementation. 

As its main indicators, the lean emergencies project uses the National Emergency Department Overcrowding Score (NEDOCS) for measuring the number of patients in the emergency room and the risk arising from this overcrowding to its users, as well as the Length of Stay Indicator (LOS) that measures the length of stay of the patient in the emergency room. These indicators are used to check progress and identify possible irregularities in the process so that these can be corrected [16], remembering that these indicators were monitored only at the beginning of the project.

Another source of data refers to the practical experience of the author who, at the time of the implementation of lean at HC-UFU, actively participated in the entire process of training, implementation, mentoring and monitoring of activities, along with all activities conducted with tutors from the Sírio-Libanês Hospital. The author, who was also part of the lean in emergencies implementation team at the Clinical Hospital (Uberlândia), actively participated in the negotiations with the Municipal Health Department of Uberlândia, the Public Prosecutor’s Office, the administration of UFU and the Sírio Libanês Hospital.

Throughout the study, there were also debates with other members who participated in the implementation of lean in emergencies at the Clinical Hospital of Uberlândia. The discussions were guided along a technical path, analyzing the scenarios with impartiality and with a broader view, considering political and cultural factors, and particularities of a university hospital, all of which were oriented toward the vision of the health system.

### 2.2. How the Implementation Process of Lean in Emergencies Occurred at HC-UFU

The steps for implementing “lean in emergencies” at HC-UFU were executed using the following schedule: in April 2018, it joined the project; on 25 and 26 May, the initial diagnosis was made; from 4 to 6 June, the formation of the working group occurred in São Paulo; on 19 June, there was a meeting with managers and leaders to present the project and agreements; on 20 and 21 June, the first visit by the consultants of Sírio Libanês Hospital took place to present the diagnosis and construct the values flowchart. On 25 June, the implementation of the 5S movement began, which originated in Japan in the late 1960s, as part of the country’s post-World War II reconstruction process. It is a philosophy that aims to mobilize all employees of an organization, through the implementation of changes in the work environment, which includes the elimination of waste.

The method is called 5S because, in Japanese, all 5 associated words start with the letter S and represent each step of this method: Seiri—organization, use and disposal; Seiton—arrangement and ordering; Seisou—cleanliness and hygiene; Seiketsu—standardization; and Shitsuke—discipline [17]. Encompassing the values of 5S, on 3 July, there was a meeting with the HC-UFU medical team to raise awareness and reach an agreement. On 20 July, the implementation of the 5W2H tool began. This tool is an action plan used to find a solution for a specific contingency of an organization, with the objective of identifying actions through the establishment of methods, deadlines and related resources.

The methodology used represents the initials of the words: What (what will be done, steps), Why (why the task should be performed, justification); When (when each task will be performed, time); Where (where each step will be performed, location); How (how it will be performed, method); How much (how much each process will cost, price); and How to measure (how it will be measured or evaluated, monitoring) [18].

On 28 September 2018, the fortnightly meetings with the consultancy ended, which went over to monthly, with a scheduled duration of monitoring by the tutors of Hospital Sírio Libanês for six months.

### 2.3. Description of the Implemented Lean System

Lean healthcare is based on a five-step process that, after the adaptation of the lean principles [19], was presented as (a) defining client value to meet their needs, such as, for example, diagnostic tests and indicated therapy; (b) flowchart map value, which includes the definition of the activities from start to finish of the process steps; (c) review the value stream to identify waste and eliminate it, that is, adapt and be efficient in health care; (d) push forward, defined by the ability to signal the pace of activities for the following stages, with a view to avoiding excessive stocks; and (e) the pursuit of perfection, a step that should drive the continuous improvement of lean healthcare with timely and quality care [5,20].

Lean thinking consists of a systematic approach that allows the identification and elimination of waste in production processes, with the focus being on adding quality and delivering to the client only what they consider to be of value. In other words, lean is the maximization of value to the client, using an efficient process and without waste. In the area of health, this means providing services that respect and respond to the preferences and needs of the patient. Another principle is the elimination of activities that do not generate value together with other waste elements (such as long waiting times for attendance, repetitive steps and conflicting advice on treatment). This waste does not allow the patient to pass through the care and treatment process without interruptions, detours, unnecessary follow-ups or waiting. Thus, with the elimination of these activities, the efficiency of actions and the quality of care are simultaneously increased [21].

## 3. Results

For this study, and in accordance with information available in the Statistics Sector of HCU-UFU, July/2018 was considered as the month/year of initiating the presentation of data for this research study. The results of this research are presented in Figure 1 and Figure 2, and in Table 1, Table 2 and Table 3, below.

The NEDOCS score calculates the saturation/overcrowding in the emergency department, where it acts as the main indicator of the project, as it lists several criteria for the definition of hospital overcrowding at any given time. It is desirable that the initiatives developed in the project result in the reduction of this indicator. An evolutionary history of NEDOCS measurement from July 2018 to May 2019 is demonstrated in the chart below.

According to the NEDOCS score, data should be collected twice a day, with the following data in the collection: the number of patients in the emergency department; service points in the emergency room, not counting the extra beds; the number of patients hospitalized and awaiting admission to the hospital (with an indication of hospitalization), that is, those who would not be indicated to remain in the emergency room; the number of effective hospital beds available for the emergency service, without considering deactivated beds, surgical blocks or beds occupied for more than 90 days; the number of patients on the ventilator; longer hospitalization time in hours, from entering the emergency room to transferring to the destination unit; and waiting time to arrive at the bed, that is, the time between information on bed availability and actual admission. As it was not possible to obtain the necessary data related to NEDOCS, for the three periods studied (T1, T2 and T3), indicators were adapted from the Statistics Sector that managed to approach a reality close to that expected in accordance with NEDOCS and length of stay (LOS).

The Length of Stay (LOS) measures the length of stay of the patient in the emergency room. Data were stored from May 2018 to May 2019 and, during this period, the length of stay of patients remained high.

During the period in which it was possible to collect data on the two indicators (NEDOCS and LOS), the motivation of the team gradually improved, despite a growing tension between municipal management and hospital management, with the involvement of the judiciary, the university administration, as well as representatives of all municipalities in the region. This tension culminated in the dismissal of the first director of HC-UFU on 12 February 2019, which coincided with the implementation cycle of lean in emergencies.

Noteworthy here is that during this period, despite the fluctuations, the length of stay of the patient in the emergency room remained high.

Indicators such as NEDOCS and LOS, which would be the main references for analyzing the implementation of the lean project in emergencies and its results, were lost during the implementation. For the elaboration of the comparative analysis, and given the scarcity of available data, we selected some indicators that were closer to that which we intended to analyze. The following indicators were defined: turnover rate or bed renewal rate; average flow/day; average patient/day; average length of stay; discharge rate and employee turnover rate over the periods T1 (twelve months before implementation), T2 (twelve months during implementation) and T3 (twelve months after implementation).

For this study, and according to the data encountered, 30 April 2018 was considered as the project start date. Information from the statistics sector, considering the daily census, was collected with authorization from the HC-UFU.

Note here that, with the exception of the variable for female deaths of less than 24 h after admission, all other variables present percentual increases before and after the intervention.

In the steps analyzed, one notes that there was a higher prevalence of female admissions when compared to those of males (617.7, 641.92, 671.50 versus 455.3, 452.92, 517.17), respectively.

In the analysis of deaths that occurred in less than 24 h and with more than 24 h after admission, there is a noted prevalence in male deaths when compared to female deaths, thus showing that in the three stages analyzed (3.1, 2.83 and 4. 00 versus 2.00, 1.08 and 2.25) for less than 24 h after admission, respectively, and (7.00, 8.67 and 12.75 versus 6.00, 7.92 and 9.67), respectively.

As for the variables, male hospitalization, female hospitalization, male transfer, female transfer, male death after 24 h and female death after 24 h, all showed significant differences between the first and third stages, demonstrating an increase in the average number of hospitalizations. There was no significant difference between stage 2 and the remaining stages. 

Male discharge and female deaths before 24 h did not present an increase in the averages for the three stages.

The variables of female transfer and female deaths in less than 24 h present a significant difference between the first and second stages. 

Trend analysis over the period shows that hospitalizations related to males had a percentage growth trend of 2.54%, while for females the growth was 1.37% (*p* < 0.058 and positive coefficient).

The variables male discharge, female discharge, male transfer and male and female death within 24 h were stationary, although these fluctuated over the years (*p* > 0.05).

However, attention is drawn to the increase in male and female deaths with less than 24 h after admittance, with elevated percentages of 11.14% and 9.45%, respectively (*p* < 0.05 and positive coefficient).

The graph in Figure 3 presents the analysis of the general occupancy of PS/HC-UFU, over the period of 36 months, between April 2017 and March 2020, with one year for stage T1 (April 2017 to March 2018), one year for stage 2, T2 (April 2018 to March 2019) and one year for phase T3 (April 2019 to March 2020).

One notes that in this graph, phase T1 started with an occupancy rate of 118.4% and ended with a rate of 141.0% in March 2018. Phase T2 had an occupancy rate of 145.8% in April 2018, dropping to 106.7% in August 2020, and then ending in March 2019 with an occupancy rate of 127.8%. Finally, stage T3 had an occupancy rate of 137.1% in April 2019 and was more pronounced in August 2019, with a total occupancy rate of 161.5%.

## 4. Discussion

The present case study, of a descriptive and exploratory nature, evaluated the results obtained through the implementation of a management system that has been implemented across various health services—lean healthcare. Theoretically, this system is a health management method that allows for the identification of points where waste occurs along a productive chain. Once identified, these points become subject to correction, which can be altered or canceled. As a result, resources can be redirected to other fronts aimed at improving the quality of services as a whole.

A modern and multifaceted hospital, that intends to be competitive, should organize its services to improve care, creating conditions favorable to its internal public and, principally, external public, extending as such its users, while optimizing its finite resources. Among the services that need to be suitably organized, the emergency department is highlighted through its complexity and scope of actions. The urgency and emergency attendances need to be planned, programmed and operationalized to meet the SUS principles, as well as improve the resolution and satisfaction of the care team and the user of the health system. However, due to structural deficiencies in the healthcare system, the emergency department services end up constituting the gateway to hospitals and represent, for the user, the possibility of accessing more resolute and complex care [23].

The application of lean in emergencies at HC-UFU did not have an impact on the indicators used in this study when compared to the pre (T1), present (T2) and subsequent (T3) periods of its implementation. According to the study by Soliman and Saurin [10], cultural barriers and practices should be overcome to reach correct and effective dissemination for the application of the lean philosophy to the health sector. The authors Kim & others [24] and D’an-dreamatteo et al. [25] state that the readiness to respond, on the part of professional health technicians, to cultural change is a positive factor and decisive in the implementation of lean healthcare at health units. However, jointly with the implementation, there was a monthly follow-up of the indicators and there was no alteration noted in their components. In fact, various factors may have influenced this result. One such factor is in relation to the complexity of the emergency attendance sector. It is known that the attendance of urgency and emergency is very complex, as it considers a number of implied factors, such as the open door policy that brings unpredictability; the need for differentiated attendance in cases of violence and accidents that are more common in urban centers; the diffusion of pre-hospital emergency care services; the process of transferring the management of the public health system over to municipalities; the multiple interests of managers and service providers; the inexperience of many managers and the expectation of comprehensive care on the part of the population. In addition to these challenging circumstances, there also exist structural fragilities in the health system. 

By analyzing the behavior of the scores intrinsic to lean in emergencies (NEDOCS and LOS), reported only during the operational period of the system, one notes that these also do not present a trend toward improvement. This demonstrates that an eventual blinding of outcome, due to the use of substitutive criteria for creating quality scores, apparently does not occur in the present study.

Independent of the difficulty in dealing with this problem, the main justification for investing in emergency services should continue to be a social and health responsibility, that is, to reach a reduction in death rates [26]. Therefore, the development and application of better management tools should be a goal of any emergency department administration. 

Another factor to be considered when analyzing the failure to obtain the desired results with the application of lean in emergencies involves the techniques employed in its implementation, the motivational climate found in the emergency room and aspects of a political-administrative nature that were ongoing at that time. During the implementation period and system implementation, the two indicators (NEDOCS and LOS) were monitored, and the teams were found to be highly motivated, although there was a growing tension between the municipal administration and hospital administration, with the active involvement of different spheres, those being judiciary, university administration and representatives of regional municipalities. This tension culminated in the dismissal of the first director of HC-UFU on 12 February 2019, hence, during the implementation cycle of lean in emergency departments. As a further aggravation, during the period of the study there was already a movement that would result in the transfer of the management of the HC-UFU from the original foundation that supported the management of the hospital, the Fundação de Assistência, Estudos e Pesquisa de Uberlândia (Assistance, Studies and Research Foundation of Uberlândia) (FAEPU), to the Empresa Brasileira de Hospital Services (Brazilian Company of Hospital Services) (EBSERH), which is a public company governed by private law, linked to the Ministry of Education, which has the sole purpose of providing medical and hospital assistance services. This situation certainly generated concern and uncertainty among hospital workers [26], mainly due to the fact that, among the objectives of EBSERH, was the replacement of FAEPU workers with candidates from EBSERH.

The present study also sought references that allowed for the comparison of the indicators from other implementation processes of the lean methodology and their results, performed in other emergency rooms outside of the HC-UFU. To this end, an extensive systematic literature review (SLR) was performed, but, unfortunately, the methodology used in the review resulted in little information being found.

In an isolated article, Soliman and Saurin [10] performed an analysis of the processes linked to the implementation of lean healthcare at five Brazilian hospitals, where the complexity of the implementation was frequently mentioned in the publication and attributed to unpredictability, along with variability of treatments and necessary tests.

The authors Costa & cols. [5] analyze five processes for the implementation of lean performed in two Brazilian hospitals; however, these did not demonstrate a detailed sequence of steps for the implementation of lean practices (LP), nor did they extract experiences that could be translated into guidelines with the potential to aid other lean. In addition, Radnor, Holweg & Waring [8] affirm that a majority of studies on the topic of lean healthcare are not comparative, as these have been developed, in most cases, using isolated case studies. In our SLR, no structured research was found that used criteria and indicators of lean in emergency departments similar to those analyzed throughout the study herein, a fact that does not allow for the comparison with other experiences external to it.

Additionally, the personal experience of the author was used in the methodology of the present research, who actively participated in the implementation process of the new management methodology. This experience is contained in many of the observations made throughout the text and contributes toward a better understanding of the study.

Only a limited number of points from these observations will be highlighted herein. As previously described, the NEDOCS and LOS indicators were lost during the implementation of lean in the emergency department at HC-UFU, largely as a result of changes to the top management of the institution and local problems that arose during the process.

Another noteworthy aspect that one can highlight was the pivotal moment in which the decision was made to participate in the project, which, as happens in nearly all cases, arose from the higher institutional leadership. It was not, therefore, a process that matured over the long term and whose need was understood and incorporated by the internal and external communities of the unit. This aspect added uncertainties, insecurities, and a lack of commitment to planning and communication, as well as to the changes that would be adopted.

These facts suggest that the complexity of implementing managerial changes, whatever they may be, is lessened when the high institutional hierarchy understands, supports and encourages the methodologies and procedures of such changes, from the outset of the knowledge awareness process to the implementation in the hospital units; it is palpable that the involvement of senior institutional and extra-institutional leadership and the commitment of employees in all processes ensures that changes are not lost [10,27].

However, longitudinal studies to monitor the long-term results of lean implementation and its consequences for institutions, teams and patients are still scarce. Systematic reviews point to the need to improve the quality of evidence, in addition to the use of clear terminology for lean healthcare, which converts the language of manufacturing to a language focused on health [21,28].

It can be assumed that the lack of success in the implementation of lean in the emergency room of HC-UFU may have occurred, in greater part, due to changes in hospital management, movements that occurred in response to the change of workers from FAEPU to EBSERH and the constant conflicts between hospital management, health managers, representatives of the judiciary and other actors involved in the process.

Finally, it should be noted that the results analyzed did not demonstrate a permanent evolution in the improvement of working conditions or the quality of care provided by the unit, facts that are mainly reflected in the behavior of the occupancy rate of the emergency service, as well as in the other indicators that were studied.

## 5. Conclusions

It is known that a structural change in an institution, such as that which is intended for the implementation of lean healthcare in the emergency room of HC-UFU, requires prior cultural and philosophical changes. This necessarily implies a lot of commitment from hospital management and other actors, such as municipal and regional managers, the public ministry, the press, representative bodies of society and the professional training academy. Due to the involvement of such bodies and organizations, there is a demand for efficient communication, a facet that is fundamental to the success of the undertaking, which in this case did not occur.

The lean in emergencies project depends on people and one of the biggest challenges is to engage these in the project and promote the necessary cultural changes. Therefore, the implementation requires both political and social stability of the hospital unit, without unrest or unexpected changes of managers, who may not agree with the ongoing changes. The intended management change naturally implies a different view of the same object, and the political and technical spaces need balance to move forward and achieve the objectives set out in the proposal.

Considering the cultural aspect of the institution, the training of physicians and other health professionals presents different parameters from those present in hospitals that do not train these professionals; aspects of these parameters certainly are present in the behavior of the analyzed indicators and need to be respected and considered in the final evaluation of management results.

There is no doubt about the need to improve management flows and processes to speed up patient discharge and provide quality care to patients who arrive at the emergency room with their varied and particular health conditions, a point worthy of note here is that we need to improve the internal service flow, which means prioritizing the best service, with greater agility, while directing the patient to the most suitable treatment point, so as not to remain in the emergency room for longer than originally calculated. 

The functioning of the emergency room with an open-door policy needs to be reassessed and, if modified, it must be replaced primarily using the experiences and achievements of other health units that have been previously planned and put into operation, so as not to cause damage to the health of users.

Communication of the process for implementing lean in emergencies was one of its weak points, both internally and externally, a fact that may have generated and aggravated resistance to the proposal and brought misunderstandings about what its final purpose would be.

The study conducted herein endorses the conclusion that the general objective of the project, which was to analyze the effectiveness of the implementation of the lean healthcare system in the emergency room of the Clinical Hospital of the Federal University of Uberlândia (HC-UFU), from the comparison of hospital indicators obtained as of the three phases that correspond to the period of one year before implementation (T1), one year during the implementation (T2) and one year after implementation (T3), was not accomplished. This is seen through the numbers obtained during the study that did not substantiate the original intention, as well as not demonstrate the success of its implementation in the emergency room of HC-UFU.

Given this aforementioned situation found by the study, a new critical look at the data presented was deemed necessary and, thus, the absence of favorable results at that time can be used to aid in improving the process. 

There is nothing better than recognizing the deficiencies and using these as positive experiences for the continuous improvement of the work, even more so considering that lean in emergencies continue to occur in other public hospital units, and this work can thus go on to facilitate in the reduction of the margin of error during the implementation of the system.

Lean in emergencies is a project sponsored by the Ministry of Health, and its objective is to reduce overcrowding in hospital emergency rooms through the lean methodology, a management philosophy for improving processes. The study demonstrates that what was expected did not happen in the experience of implementing lean in the emergency department at HC-UFU. Therefore, it is believed that the experience was not successful; however, the results of this research will favor a redirection of activities towards organized and efficient care of the population and new behaviors to reduce the waste of resources in the activities of nurses and doctors.

Therefore, the conclusion is reached that this study presented the difficulties encountered by HC-UFU in implementing the lean project in the emergency department, thus ensuring that other institutions that intend to implement this project do not make the same types of mistakes.

## Figures and Tables

**Figure 1 ijerph-20-07184-f001:**
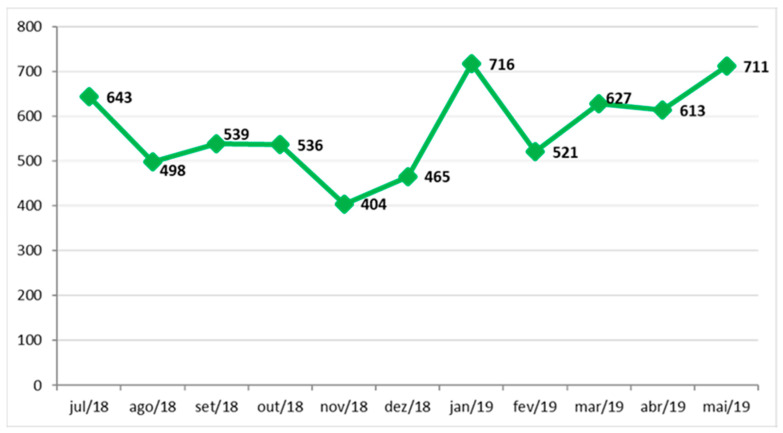
Evolutionary history of NEDOCS measurement. Source: Hospital Information System (SIH), [22].

**Figure 2 ijerph-20-07184-f002:**
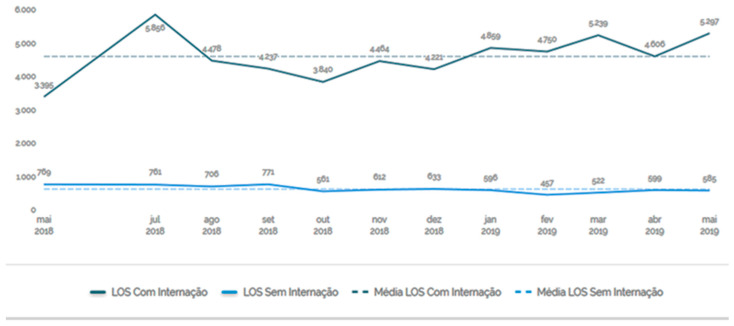
Length of stay: length of stay of the patient in the emergency room. Source: Hospital Information System (SIH), [22].

**Figure 3 ijerph-20-07184-f003:**
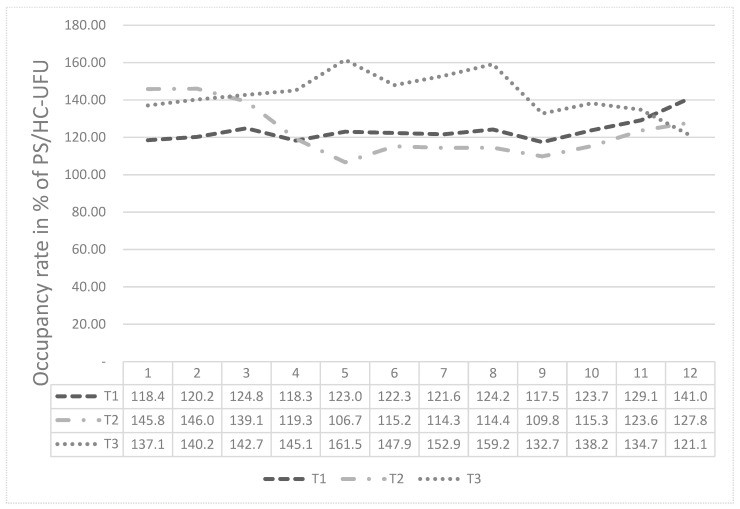
Analysis of general occupancy rate of the PS/HC-UFU. Uberlandia, MG. Source: Hospital Information System (SIH) [22].

**Table 1 ijerph-20-07184-t001:** Relative and absolute frequency of attendances performed at the PS/HC-UFU in the Pre, Intra and Post-intervention Stages.

	T1		T2		T3		
	f	%	F	%	F	%	Total
Male hospitalization	5463	31.9	5435	31.8	6206	36.3	17,104
Female hospitalization	7412	32.0	7703	33.2	8058	34.8	23,173
Discharge male	3085	33.8	2617	28.7	3419	37.5	9121
Discharge female	2732	34.1	2593	32.3	2692	33.6	8017
Male transfer	2248	29.8	2676	35.4	2625	34.8	7549
Female transfer	4607	31.0	5005	33.7	5226	35.2	14,838
Male death < 24 h	37	31.1	34	28.6	48	40.3	119
Female death < 24 h	27	40.3	13	19.4	27	40.3	67
Male death > 24 h	88	25.5	104	30.1	153	44.3	345
Female death > 24 h	66	23.8	95	34.3	116	41.9	277

Source: from the authors, 2021.

**Table 2 ijerph-20-07184-t002:** Comparison between the averages of the various indicators found in the three stages before, during and after the intervention with the implementation of lean healthcare carried out at the PS/HC-UFU. Uberlândia, MG, Brazil. N (40.277).

Phases		Average	DP			
				CI95%	CI95%	*p*-Value
	1	455.25	31.73	435.09	475.41	<0.001 *
Male hospitalization	2	452.92	35.93	430.09	475.75	0.988
	3	517.17	44.76	488.73	545.60	<0.001 *
	1	617.67	38.63	593.12	642.21	0.013 *
Female hospitalization	2	641.92	35.51	619.36	664.48	0.211
	3	671.50	50.46	639.44	703.56	0.013 *
	1	257.08	17.22	246.14	268.03	<0.001 *
Male discharge	2	218.08	26.70	201.12	235.05	<0.001 *
	3	284.92	23.26	270.14	299.70	<0.001 *
	1	227.67	22.26	213.53	241.81	0.481
Female discharge	2	216.08	20.30	203.18	228.98	NC
	3	224.33	28.37	206.31	242.36	NC
	1	187.33	19.25	175.10	199.56	0.022 *
Male transfer	2	223.00	42.49	196.01	249.99	0.941
	3	218.75	27.27	201.42	236.08	0.049 *
	1	383.92	23.92	368.72	399.12	<0.001 *
Female transfer	2	417.08	19.10	404.94	429.22	0.152
	3	435.50	27.11	418.27	452.73	0.042 *
	1	3.08	1.83	1.92	4.25	0.279
Male death < 24 h	2	2.83	1.75	1.72	3.94	NC
	3	4.00	1.95	2.76	5.24	NC
	1	2.25	1.29	1.43	3.07	0.013 *
Female death < 24 h	2	1.08	0.29	0.90	1.27	0.006 *
	3	2.25	0.97	1.64	2.86	0.963
	1	7.33	2.23	5.92	8.75	0.001 *
Male death > 24 h	2	8.67	2.74	6.92	10.41	0.587
	3	12.75	4.47	9.91	15.59	0.001 *
	1	5.50	1.68	4.43	6.57	0.023 *
Female death > 24 h	2	7.92	4.21	5.24	10.59	0.449
	3	9.67	4.05	7.09	12.24	0.017 *

* ANOVA test; steps: 1—Pre-intervention, 2—Intra-intervention, 3—Post-intervention, NC—*p* value not calculated. Source: from the authors, 2021.

**Table 3 ijerph-20-07184-t003:** Trend in rates of consultations performed at PS/HC/UFU. Uberlândia, MG, Brazil, 2017–2020. N = (40,277).

	β	CI 95%	CI 95%	*p*-Value	APC	CI 95%	Interpretation
Male hospitalization	0.0251	0.0109	0.0394	0.0010	2.5461	(1.14/3.97)	ascendant
Female hospitalization	0.0136	−0.0001	0.0273	0.0511	1.3699	(0.04/2.72)	stationary
Male discharge	0.0283	−0.0081	0.0646	0.1234	2.8679	(−0.68/6.54)	stationary
Female discharge	−0.0015	−0.0254	0.0224	0.8990	−0.1505	(−2.43/2.18)	stationary
Male transfer	0.0107	−0.0286	0.0499	0.5849	1.0723	(−2.69/4.98)	stationary
Female transfer	0.0171	0.0040	0.0303	0.0121	1.7296	(0.4/3.03)	ascendant
Male death < 24 h	0.0553	−0.0634	0.1739	0.3507	5.6822	(−5.7/18.51)	stationary
Female death < 24 h	0.0435	−0.0521	0.1390	0.3622	4.4412	(−4.76/14.53)	stationary
Male death >24 h	0.1056	0.0585	0.1528	0.0001	11.1427	(6.20/16.32)	ascendant
Female death > 24 h	0.0904	0.0052	0.1755	0.0382	9.4557	(0.82/18.83)	ascendant

Prais–Winsten regression analysis. Source: from the authors, 2021.

## Data Availability

Data are contained within the article.

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
