# Peer review of "Implementation of the Lean Healthcare System in the Emergency Room of the Clinical Hospital of the Federal University of Uberlândia: A Case Study"

_ijerph, 2023, doi:10.3390/ijerph20247184_

Round 1

Reviewer 1 Report (Previous Reviewer 1)

Comments and Suggestions for Authors

There are two figures between lines 449 to 452 identified as figure 2. I believe that one is the one that was erased to rectify the font size, but it was still indexed to the text. Remove the figure and increase the intensity of the font colors for better reading in the original figures.

Author Response

The figure has been removed. Formatted font color intensities.

Reviewer 2 Report (Previous Reviewer 3)

Comments and Suggestions for Authors

Thank you for the revision and resubmission.

Author Response

Thanks for the comments and suggestions.

Reviewer 3 Report (New Reviewer)

Comments and Suggestions for Authors

The authors investigated the possible effects of applying 5S methodology, one of the lean production techniques, in an emergency room to overcome the administrative difficulties encountered in emergency departments. Emergency services and healthcare systems are faced with worsening performance every year as a result of increasing population and limited resources. Lean production techniques, on the other hand, eliminate waste sources in businesses and increase the performance of the systems. In this context, in recent years, lean production techniques have begun to be applied in healthcare systems. This study is important at this point. However, the following issues must be addressed before publication:

1. The statement "hospital indicators" is mentioned in the abstract. Some indicators should be given in parentheses.
2. How three phases are determined should be explained in the text.
3. Abbreviations should not be used in the abstract (i.e., NEDOCS and LOS).
4. Implementation of lean production enables activities within the companies to be carried out faster and reduces the waste associated with the waiting times of operational activities. Therefore, no improvement may be expected in indicators related to patients, but a decrease in wasteful resources in the activities of nurses and doctors can be expected. In other words, healthcare providers such as nurses and doctors can be enabled to work more efficiently. Therefore, a definite statement should not be used in the last sentence of the abstract.
5. What was taken into consideration when selecting 10 hospital indicators? Was literature used or did suggestions come from stakeholders during the project process?
6. Has data cleaning been performed during the data analysis process?
7. In Conclusion section of the study, it should be discussed what kind of contribution this study can make to future studies.
8. In Conclusion section of the study, the contribution of this study to relevant stakeholders (such as patients, nurses and doctors, hospital administrators, the country's health system) should be discussed.

Author Response

  1. The statement "hospital indicators" is mentioned in the abstract. Some indicators should be given in parentheses.

Authors' response: The authors chose to maintain the indicators “National Emergency Department Overcrowding Score and “Length of Stay Indicator, described in the abstract, without using parentheses.

  1. How three phases are determined should be explained in the text.

Authors' response: The three phases correspond to the periods of one year before implementation (T1), the year during implementation (T2) and one year after implementation (T3). These three phases are described within the text.

  1. Abbreviations should not be used in the abstract (i.e., NEDOCS and LOS).

Authors' response: Correction made to the text. Highlighted in Red.

  1. Implementation of lean production enables activities within the companies to be carried out faster and reduces the waste associated with the waiting times of operational activities. Therefore, no improvement may be expected in indicators related to patients, but a decrease in wasteful resources in the activities of nurses and doctors can be expected. In other words, healthcare providers such as nurses and doctors can be enabled to work more efficiently. Therefore, a definite statement should not be used in the last sentence of the abstract.

Authors' response: Rewritten the conclusion, with the following text. Highlighted in Red.

Lean in Emergencies is a project sponsored by the Ministry of Health, and its objective is to reduce overcrowding in hospital emergency rooms through the Lean methodology, a management philosophy for improving processes. The study demonstrates that what was expected did not happen in the experience of implementing Lean in Emergencies at HC.UFU. Therefore, it is believed that the experience was not successful, however, the results of this research will favor a redirection of activities towards organized and efficient care of the population and new behaviors to reduce the waste of resources in the activities of nurses and doctors.

  1. What was taken into consideration when selecting 10 hospital indicators? Was literature used or did suggestions come from stakeholders during the project process?

Authors' response: Included the following text:

With the loss of the main indicators used by Hospital Sírio Libanês, which led the implementation, it was necessary to search the HC-UFU statistics department for indicators similar to those recommended by the Lean Project in Emergencies of the Ministry of Health/Hospital Sírio Libanês.

  1. Has data cleaning been performed during the data analysis process?

Author's response: Yes

  1. In Conclusion section of the study, it should be discussed what kind of contribution this study can make to future studies.

Authors' response: Included the following text:

It is concluded that this study presented the difficulties encountered by HC-UFU in implementing the Lean Project in Emergencies, thus ensuring that other institutions that intend to implement this project do not make the same types of mistakes.

  1. In Conclusion section of the study, the contribution of this study to relevant stakeholders (such as patients, nurses and doctors, hospital administrators, the country's health system) should be discussed.

Authors' response: Included the following text:

Lean in Emergencies is a project sponsored by the Ministry of Health, and its objective is to reduce overcrowding in hospital emergency rooms through the Lean methodology, a management philosophy for improving processes. The study demonstrates that what was expected did not happen in the experience of implementing Lean in Emergencies at HC.UFU. Therefore, it is believed that the experience was not successful, however, the results of this research will favor a redirection of activities towards organized and efficient care of the population and new behaviors to reduce the waste of resources in the activities of nurses and doctors.

This manuscript is a resubmission of an earlier submission. The following is a list of the peer review reports and author responses from that submission.

Round 1

Reviewer 1 Report

Comments and Suggestions for Authors

The article is interesting and contributes to expanding research on the subject, both academically and for hospitals in general.

However, there are some necessary adjustments:

The abstract is well structured, but the texts are truncated, as for the divided categories. In the objective of the abstract, the text differs from the stated objective between lines 84 to 88 of the text. In the abstract it states "To describe the methodology used and the results found in the implementation", while in the objective of the introduction it states "The present study aimed to analyze the effectiveness of the implementation of the...".

I recommend a complete adjustment in the Abstract, keeping the same texts as the work, considering that the Abstract is not a complement, but a quick description of all the content, therefore, the same texts can be cut and inserted in the construction of the abstract, considering the categories used by the authors. The summary should also be revised in the translation into English, especially in the excessive use of pronouns, traditionally present in the Portuguese language.

I also recommend an assessment of the concordance of the entire translated text of the article.

About the Method, section 2, I recommend the elaboration of a figure that illustrates the flowchart of the research methodological procedure, not only about the techniques for implementing Lean Healthcare.

Lean is a project development technique for improving different processes, reducing waste and with the combination of 6 sigma, it is very interesting, the results obtained, with extremely satisfactory performance, but the way of implementing the "Lean tools" should not be considered as the research method, but rather as part of it, primarily for implementation.

I recommend the description of the methodological procedure in a flow (section 2.4) that describes the literature review (perhaps the PRISMA technique) better explains the selection of articles for the theoretical foundation, then the criteria for choosing the areas and "tools" to be used applied and then the sequence of activities planned to implement the "tools".

The conclusion should add a more assertive paragraph about meeting the objective proposed in the introduction of the article.

Author Response

The abstract is well structured, but the texts are truncated in terms of divided categories. In the objective of the abstract, the text differs from the stated objective between lines 84 to 88 of the text. The summary states "Describe the methodology used and the results found in the implementation", while the objective of the introduction states "The present study aimed to analyze the effectiveness of the implementation of...".

Answer: Correction was made to the purpose of the Abstract, in line with the text

I recommend a complete adjustment in the Abstract, keeping the same texts of the work, considering that the Abstract is not a complement, but a quick description of all the content, therefore, the same texts can be cut and inserted in the construction of the abstract, considering the categories used by the authors. The summary should also be reviewed in the translation into English, especially in the excessive use of pronouns, traditionally present in the Portuguese language. I also recommend a concordance assessment of the entire translated text of the article.

Answer: A complete adjustment was made to the Summary. Revised English translation. Assessment of the agreement of the entire translated text was carried out.

About the Method, section 2, I recommend the elaboration of a figure that illustrates the flowchart of the research methodological procedure, not only about the Lean Healthcare implementation techniques.

Answer: The figure recommended by the reviewer was not made. We chose to keep the text written, descriptive, following the pattern we adopted in the presentation of methods.

Lean is a project development technique for improving various processes, reducing waste and with the combination of 6 sigma, it is very interesting, the results obtained, with extremely satisfactory performance, however the way of implementing the "Lean tools" should not be considered as the research method, but rather as part of it, mainly for implementation.

I recommend the description of the methodological procedure in a flow (section 2.4) that describes the literature review (perhaps the PRISMA technique) better explains the selection of articles for the theoretical foundation, then the criteria for choosing the areas and “tools” to be used applied and then the sequence of activities planned to implement the "tools".

Answer: The flow figure recommended by the reviewer was not made. We chose to keep the text written, descriptive, following the pattern we adopted in the presentation of methods.

The conclusion should add a more assertive paragraph about meeting the objective proposed in the introduction of the article.

Answer: A more assertive paragraph was written about meeting the proposed objective.

Reviewer 2 Report

Comments and Suggestions for Authors

Abstract: The abstract of the paper is too long, exceeding the Authors Guidelines by more than twice the recommended length. Within the abstract, only the main information on the issues of the paper should be included, selecting what needs to be highlighted most.

Introduction: The introduction completely lacks a theoretical background, it is not enough to explain the situation in Brazilian public health: on what theoretical assumptions is the Lean in Emergencies protocol based? Why is it valid compared to other protocols? How was it chosen? How does it work? For example, in line 78 the authors write "Success stories are described in the literature" but cite few articles. Are these evidence-based protocols? 

MethodWhy is paragraph 2.1. in the method? Much of the information could be part of the introduction.

Results: The tables should be revised in formatting, they contain typos.

Discussion and Conclusion: The discussion section clearly summarises the findings, however, it again lacks any reference to the theoretical assumptions that led to the implementation of this model.

Minor comments:
I suggest checking the formatting.
Overall, I'd suggest an English native-speaker revision.
I would suggest implementing the bibliography consistently.
I hope that my comments are constructive and helpful.

Author Response

Abstract: The article abstract is too long, exceeding the Guidelines for Authors by more than twice the recommended length. Within the abstract, only the main information about the subjects of the work should be included, selecting what most needs to be highlighted.

Response: Rewritten the abstract in accordance with the journal's Guidelines for Authors.

Introduction: The introduction completely lacks a theoretical basis, it is not enough to explain the situation of Brazilian public health: on what theoretical assumptions is the Lean protocol in Emergencies based? Why is it valid compared to other protocols? How was it chosen? How it works? For example, on line 78, the authors write "Success stories are described in the literature", but cite few articles. Are these protocols evidence-based?

Answer: Adjusted the introduction.

Method : Why paragraph 2.1. Full name? Much of the information could be part of the introduction.

Answer: Corrections were made to the Method.

Results: The tables must be revised in terms of formatting, as they contain typing errors.

Discussion and Conclusion: The discussion section clearly summarizes the results, however it again lacks any reference to the theoretical assumptions that led to the implementation of this model.

Minor comments:

I suggest checking the formatting.

Overall I would suggest a review from a native English speaker.

I would suggest implementing the bibliography consistently.

I hope my comments are constructive and helpful.

Answer: Checked and corrected the formatting of tables and text. A new translation of the article into English was carried out by a specialized professional.

Reviewer 3 Report

Comments and Suggestions for Authors

Thank you very much for this interesting article describing the implementation, its obstacles, encountered issues and results of a Lean Healthcare system in a university emergency department. 

Many aspects resonated with my experience, while reviewing in-between supervising shifts in a comparable structure.

The context of implementation is well described. 

The material and method section could be improved in order to make it better reproductible/understandable (see below) especially considering the relevant issue encountered (which - as you explain the discussion - could be part of the explanation why the implementation did not show the desired improvement)

The discussion puts the results into a broader context.

Here my suggestions to make this article more salient to the readers and to future lean-project-leaders:

2. Material and Methods:

Especially in the context of lost/incomplete data, it would be very important to explain more clearly / in a different structure

- which variables you chose

- when you planned to collect them

- what you got

- what you had to replace by proxies

Maybe a table containing the 2.3. schedule with the indicators could help understand the project.

2.3: Line 197: explaining 5S could better fit in the section explaining other Lean elements

Line 237: since this paper does not address this problem neither (the long-term effects), this sentence should be in the discussion

Results: First paragraph (except 1. sentence) would better fit in introduction

Line 252 onwards: should be in 2.2. indicators 

Line 285 "the teams' motivation was on the rise": it means the motivation was good/high/improving, despite the increasing tension? This is what I understand from what you explain later in line 400ff, but maybe rewriting this sentence would avoid missunderstanding.

Figure 1: 30.04.2018 is considered the start, so why does the graph only start in August 2018?* Please correct the month to English 

* in line 278 you explain that you didn't have all the data due to the explained issue with management/direction changes, but used adapted indicators.

Figure 3: instead of 1-12 for the month April-March, changing the labelling would make it easier to understand the text (and also would make it unnecessary to add a legend to these numbers)

Typos/errors

line 139: "with" 

line 278: no "," after NEDOCS

line 279: "that managed"   or other correction needed to make the sentence correct

line 352: 2020 missing

line 356: August 2019 (and not 2020) I assume

Author Response

Thank you very much for this interesting article describing the implementation, its obstacles, problems encountered and results of a Lean Healthcare system in a university emergency department.

Many aspects resonated with my experience in reviewing intermediate supervisory shifts in a comparable structure.

The implementation context is well described.

The material and method section could be improved to make it more reproducible/understandable (see below), especially considering the relevant problem encountered (which - as you explain in the discussion - may be part of the explanation as to why the implementation did not show the desired result). improvement)

The discussion places the results in a broader context.

Here are my suggestions for making this article more relevant to readers and future lean project leaders:

  1. Material and Methods:

Especially in the context of missing/incomplete data it would be very important to explain more clearly/in a different structure

- which variables you chose

- when you planned to collect them

- what's wrong with you

- what you had to replace with proxies

Answer: The item Material and methods was adapted/rewritten.

Perhaps a table containing 2.3. schedule with the indicators can help to understand the project.

Answer: A schedule table was not prepared, recommended by the reviewer. We chose to keep the text written, descriptive, following the pattern we adopted in the presentation of methods.

2.3: Line 197: Explaining 5S could better fit in the section explaining other Lean elements

Answer: No changes were made. We chose to keep the 5S explanation in item “2.2. How did the implementation process of Lean in Emergencies at HC-UFU occur”, as we understand that it is more appropriate for describing the method initially implemented at HC-UFU.

Line 237: Since this article also doesn't address this issue (the long-term effects), this sentence should be in the discussion

Answer: Made suitability. Added paragraph to Discussion

Results: first paragraph (except 1st sentence) would fit better in the introduction

  Answer: Made suitability. Added paragraph in Introduction

Line 252 onwards: must be in 2.2. indicators

Answer: Made suitability. Paragraph included in Item “2.1 The search for hospital indicators”

Line 285 "teams' motivation was on the rise": does this mean that motivation was good/high/improving despite increasing tension? This is what I understand from what you explain later on line 400ff, but maybe rewriting this sentence will avoid misunderstandings.

Answer: Text has been rewritten.

Figure 1: 2018.04.30 is considered the start, so why does the chart only start in August 2018?* Please correct the month for English

* on line 278 you explain that you did not have all the data due to the explained problem with changes in management/direction, but you used adapted indicators.

Answer: Text has been rewritten.

Figure 3: Instead of 1-12 for the month April-March, changing the labeling would make the text easier to understand (and also make it unnecessary to add a caption to these numbers)

Typos/errors

line 139: "with"

line 278: no "," after NEDOCS

line 279: "who got it" or other correction needed to make the sentence correct

line 352: 202 0 0 missing

line 356: August 2019 (not 2020) I presume

Answer: The entire text has been revised into English. Made typo/error corrections.